# Foggia Prostate Cancer Risk Calculator 2.0: A Novel Risk Calculator including MRI and Bladder Outlet Obstruction Parameters to Reduce Unnecessary Biopsies

**DOI:** 10.3390/ijms24032449

**Published:** 2023-01-26

**Authors:** Ugo Giovanni Falagario, Gian Maria Busetto, Marco Recchia, Edoardo Tocci, Oscar Selvaggio, Antonella Ninivaggi, Paola Milillo, Luca Macarini, Francesca Sanguedolce, Vito Mancini, Pasquale Annese, Carlo Bettocchi, Giuseppe Carrieri, Luigi Cormio

**Affiliations:** 1Department of Urology and Organ Transplantation, University of Foggia, 71122 Foggia, Italy; 2Department of Radiology, University of Foggia, 71122 Foggia, Italy; 3Department of Pathology, University of Foggia, 71122 Foggia, Italy; 4Department of Urology, Bonomo Teaching Hospital, 76123 Andria, Italy

**Keywords:** prostate cancer, risk calculator, PIRADS, bladder outlet obstruction

## Abstract

Risk calculator (RC) combining PSA with other clinical information can help to better select patients at risk of prostate cancer (PCa) for prostate biopsy. The present study aimed to develop a new Pca RC, including MRI and bladder outlet obstruction parameters (BOOP). The ability of these parameters in predicting PCa and clinically significant PCa (csPCa: ISUP GG ≥ 2) was assessed by binary logistic regression. A total of 728 patients were included from two institutions. Of these, 395 (54.3%) had negative biopsies and 161 (22.11%) and 172 (23.6%) had a diagnosis of ISUP GG1 PCa and csPCa. The two RC ultimately included age, PSA, DRE, prostate volume (pVol), post-voided residual urinary volume (PVR), and PIRADS score. Regarding BOOP, higher prostate volumes (csPCa: OR 0.98, CI 0.97,0.99) and PVR ≥ 50 mL (csPCa: OR 0.27, CI 0.15, 0.47) were protective factors for the diagnosis of any PCa and csPCa. AUCs after internal validation were 0.78 (0.75, 0.82) and 0.82 (0.79, 0.86), respectively. Finally, decision curves analysis demonstrated higher benefit compared to the first-generation calculator and MRI alone. These novel RC based on MRI and BOOP may help to better select patient for prostate biopsy after prostate MRI.

## 1. Introduction

Prostate biopsy is the gold standard for diagnosis of prostate cancer, but it has a low diagnostic yield. In current clinical practice, the detection rate of cancer at an initial biopsy related to high PSA levels and suspected digital-rectal exploration (DRE) is approximately 40% [1], dropping to approximately 25% in the setting of screening programs, i.e., patients with serum PSA between 2.5 and 10 ng/mL [2].

In order to avoid unnecessary biopsies and reduce the overdiagnosis and over treatment of low-grade prostate cancer (PCa), EAU guidelines suggest offering further risk-assessment to asymptomatic men (defined by negative DRE and PSA level between 2–10 ng/mL) before performing biopsy, consisting of: risk calculators, imaging, and additional serums and/or urine-based tests [3].

Risk calculators are freely available online tools that, combining several clinical parameters, determine the risk of being diagnosed with prostate cancer at biopsy. 

In the last 20 years, to improve the diagnostic yield of biopsy, several predictive models have been built combining PSA and DRE results with other clinical information such as age, prostate volume (PVol), PSA % free, as well as the use of new biomarker [4].

Additionally, we previously demonstrated that bladder outlet obstruction parameters increase the accuracy of models based on simple standard parameters in the prediction of PCa and csPCa at an initial biopsy evaluation [5,6]. Based on these findings, we developed and externally validated the Foggia Prostate Cancer Risk Calculator (FPC-RC) [7,8], including variables such as covariates age, PSA, transrectal ultrasound (TRUS) prostate volume, digito rectal examination (DRE) findings and post void residual volume (PVR).

With the advent of multiparametric Magnetic Resonance Imaging (mpMRI) in the clinical practice as a reliable tool in localizing specific regions of the prostate highly suspicious for csPCa, several studies have incorporated mpMRI-related parameters into risk calculators [9,10,11,12] improving the diagnostic accuracy of each of these calculators [13]. 

The aim of the present study was to develop a novel RC “the Foggia Prostate Cancer Risk Calculator 2.0”, combining parameters used in the previous calculator with mpMRI suspicion score and MRI prostate volume, to provide a simple and comprehensive predictive model that directs the patient toward the correct diagnosis of PCa and csPCa, identifying with as little doubt as possible the lesion of interest without incurring in unnecessary biopsy samples. 

## 2. Results

### 2.1. Study Population

Between January 2018 and May 2021, a total of 728 patients underwent TRUS-guided PBx at the two study centers. Of these, 395 (54.3%) patients had negative biopsy, 161 (22.11%) had a diagnosis of ISUP GG 1 PCa, while 172 (23.6%) were found to have clinically significant PCa (ISUP GG ≥ 2). Descriptive characteristics of the study population are presented in Table 1.

Patients with PCa were older (66 and 69 years, respectively, for ISUP GG 1 and ISUP GG ≥ 2 vs. 65 years old patients having negative biopsy; *p* value: < 0.0001), had higher rates of suspicious DRE (31.5% and 55.2% for ISUP GG 1 and ISUP GG ≥ 2 vs. 28.9% negative), higher PSA levels, and lower Prostate volume. 

Patients with negative biopsies had worse symptoms (LUTS, IPSS) and bladder outlet obstruction parameters in terms of PVR and Qmax. 

Regarding MRI, cancer detection rates according to PIRADS score are presented in Appendix A. Among patients having negative MRI (PIRADS 1-2), 19 (13.57%) had a diagnosis of low-grade tumor and 4 (2.85%) of high-grade tumor. In patients with PIRADS 3, 40 (21.6%) had a diagnosis of low-grade PCa and 27 (14.6%) of csPCa. Of patients with PIRADS 4, 86 patients (28.9%) resulted with a diagnosis of low-grade cancer, while it was 89 patients (29.7%) of csPCa. Among patients with PIRADS 5, 17 (15.88%) had a diagnosis of low-grade cancer and 52 (48.59%) of csPCa.

### 2.2. Multivariable Analysis and Statistical Model Development

The results of multivariable analysis are presented in Table 2 and Appendix A. All the covariates included in the models were predictors of any PCa and csPCa. Regarding bladder outlet obstruction parameters, both higher prostate volumes (any PCa: OR 0.98, CI 0.97, 0.98; csPCa: OR 0.98, CI 0.97, 0.99) and PVR ≥ 50 mL (any PCa: OR 0.57, CI 0.39, 0.84; csPCa: OR 0.27, CI 0.15, 0.47) were protective factors for the diagnosis of any PCa and csPCa. The developed model was used to create a nomogram to graphically compute the predicted probability of csPCa (GG ≥ 2 PCa) using the Foggia Prostate Cancer Risk Calculator 2.0 (Figure 1).

### 2.3. Model Performance and Internal Validation

Model calibration was graphically evaluated (Figure 2) with minimal variation between the observed and predicted probabilities for both the PCa and the csPCa predictive model.

Furthermore, the model demonstrated consistency even during internal validation with only a slight lowering of AUC values following LOOCV; in the case of PCa with any GG the AUC was 0.78 (0.75, 0.82), whereas for csPCa it was 0.82 (0.79, 0.86). Finally, decision curve analysis has demonstrated a distinct benefit associated with the use of model derived probabilities to decide whether to perform or not to perform prostate biopsy. In fact, the analysis has shown that the model predicting PCa has a clinical benefit superior to the first-generation calculator and MRI alone for each probability cut-off higher than 15%, while the one predicting csPCa at every probability cut-off lower (Figure 3).

## 3. Discussion

The current guidelines of the European Association of Urology advise against following a single model of risk interpretation in the early detection of PCa and emphasize the importance of patient participation in the decision-making process when indication for biopsy is included [3].

Using clinical and instrumental parameters that are widely used for initial urological evaluation of men at risk of PCa, we develop a new tool capable of calculating the individualized risk of PCa and csPCa in each patient.

The addition of PIRADS scores to the data offered by the previous calculator (age, PSA, DRE, PVol, PVR) [7] increased the predictivity of the model from 0.76 to 0.78 for any GG of PCa and from 0.80 to 0.82 for csPCa. Although MRI provides key findings, these by themselves do not allow us to diagnose PCa with certainty. Indeed, up to 50% of patients with a positive MRI do not have PCa at a subsequent biopsy [14,15] and tools to improve the positive predictive value of PIRADS scoring system [16] as well as risk calculators or biomarkers to stratify patients before and after MRI are needed [17,18]. In this scenario, the FPC-RC in its first version without MRI findings [7] may be used to risk stratify patients before MRI while this new version, including MRI, will help in the risk stratification after MRI [19]. Moreover, the new RC includes MRI computed prostate volume and biopsy setting as covariate. Prostate volume estimates using MRI have been shown to be similar but slightly more accurate compared to prostate volume estimates using TRUS, while the risk of being diagnosed with PCa after a previous negative biopsy is lower compared to the risk of PCa in biopsy naïve patients (OR: 0.40, IQR: 0.27, 0.58). As a result, the AUC, the calibration, and decision curve analysis demonstrated that the new calculator provides a distinct benefit in predicting the presence of csPCa compared to the previous model. 

Compared to previously published models [9,10,11,12], our new calculator includes bladder outlet obstruction parameters in addition to MRI. This indeed increases its ability to differentiate a malignant lesion from a benign one in men with lower urinary tract symptoms (LUTS). In fact, we previously showed that there is an inverse correlation between PVol and risk of harboring PCa in men who will undergo biopsy [6,20] and usually, these men present with higher PSA values due to higher grade of intraprostatic inflammation. We recently showed that PSA density rather than PSA should be used to rule out the risk of high PSA due to inflammation or LUTS [21]. Our risk calculator has been implemented both of these parameters. In addition, we tested two models in the development phase, one with PSA density and one with PSA and Volume as separate covariates. The latter showed the highest AUC and best calibration and in both the models, PVR showed independent additive predictive value. We believe this is an extremely important point to support the use of RCs compared to PSA density and PIRADS alone.

In our model we have different beta coefficients of the logit function for PSA (positive value) and prostate volume (negative value). This allows the achievement of a better prediction of PCa risk. Conversely, using their ratio (i.e., PSA density) could be considered as an excessive simplification since PSA density does not have a linear effect on PCa prediction. Indeed, even if the peripheral zone contains a significantly greater area density and absolute volume of epithelium than the transition zone, the serum PSA level is most strongly correlated with the volume of epithelium in the transition zone [22]. While we are not able to perform analysis on the transition/central zone volume since these were not included in our data collection form, we believe that further risk calculators should include them to further improve discrimination between patients at risk of PCa and patients with BPH. 

MRI represents, on the other side, a method of considerable importance to avoid prostate biopsy and to locate within the prostate the areas of interest. The accuracy of prostate sampling is indeed higher in patients undergoing a combination of systematic and target biopsy [23]. Strengths of the study include its prospective nature, the large sample size from two institutions, the availability of UFM and PVR for all the included patients and the use of an extensive (≥18 cores per patients) and standardized scheme for prostate sampling. Potential limitations are related to the fact that it is a two-site study that includes only Caucasian men. Even if previous studies have shown similar accuracy of MRI in different populations [24], external validation of our findings is needed. 

## 4. Materials and Methods

### 4.1. Study Population

All patients undergoing prostate biopsies with a pre-biopsy mpMRI at the two study centers were included in the present study. Patients were selected for prostate mpMRI because of increased serum PSA levels (≥3 ng/mL) and/or abnormal DRE. Patients who underwent invasive treatments for benign prostatic hyperplasia (BPH), patients with a previous positive biopsy, patients with foley catheter, and patients with a PSA > 20 ng/mL were excluded from the present study. The study protocol was approved by the Ethics Committee of the University Hospital “Ospedali Riuniti”, Foggia, Italy, and since this was a retrospective study, written informed consent for research was waived. All of the study participants provided written informed consent for the diagnostic procedure.

### 4.2. Mp-MRI of the Prostate

MpMRI was performed at two tertiary care academic hospitals (Ospedale Bonomo, Andria and University of Foggia) and interpreted according to the PI-RADS v2.1 criteria by two dedicated uro-radiologists with 10 years of experience in prostate mpMRI. The MRI involved the application of a 1.5 T scanner and surface array coils and/or with an end fire coil combined with a 16-channel surface coil. The sequences acquired were: i. T2-weighted morphological images (axial, sagittal, and coronal); ii. diffusion-weighted images (DWI/ADC); iii. dynamic perfusion-weighted images (DCE) through injection of 1 mL/kg gadobutrol followed by 20 mL of saline solution using an automated injector at a speed of 2 mL/s. MRI prostate volume was used in the models and to compute PSA density.

### 4.3. Bladder Outlet Obstruction Evaluation and Prostate Biopsy

According to our biopsy protocol, all patients underwent serum PSA testing, DRE and TRUS. Uroflowmetry (UFM) with evaluation of PVR was performed before prostate biopsy. All the procedures were performed in the outpatient clinic under non infiltrative anesthesia [25]. Patients who tested negative on MRI (PI-RADS 1 and 2) underwent standard transrectal ultrasound guidance (TRUS) guided biopsy according to our 18 cores template. Those who, instead, presented suspicious areas on mpMRI underwent standard and target sampling (2 to 4 additional cores from each suspicious lesion) using an electromagnetically tracked MR/ TRUS Fusion system [26]. Biopsy cores were evaluated by a dedicated uro-pathologist, who assessed International Society of Urologic Pathology Gleason Grade groups (ISUP GG) [27].

### 4.4. Statistical Analysis

Descriptive statistics of the overall cohort and according to biopsy results was performed. Continuous variables are reported as medians and confidence intervals and compared by the Wilcoxon Mann–Whitney test for independent groups. Categorical variables are reported as percentages and compared by the chi-square test. The value of the different clinical variables in predicting PCa (ISUP GG ≥ 1) and csPCa (ISUP GG ≥ 2) was assessed by multivariable binary logistic regression analysis. For the selection of variables to be included in the models, we relied on the PIRADS score (12 vs. 3 vs. 4 vs. 5) in addition to the variables we included in our previous model (Age, DRE, biopsy history, PSA, Prostate volume and PVR) [7]. 

Two risk calculators were then built based on the coefficients of the logit function. Finally, leave-one-out cross-validation (LOOCV) was performed. Calibration was graphically assessed, the area under the curve (AUC) and decision curve analysis (DCA) were used to evaluate the performance of the model using the probability derived from the risk-calculator with the coefficients adjusted after internal validation. Statistical analyses were performed using STATA 16 (StataCorp LP, College Station, TX, USA). All tests had a significance level set at *p* < 0.05.

## 5. Conclusions

The present study demonstrated that the addition of the parameters of MRI and bladder outlet obstruction (BOO) to standard clinical variables improves the predictive accuracy of risk calculators for prostate cancer. This new risk calculator represents a free, simple, and useful tool in order to better estimate the risk of PCa diagnosis and consequently guide the correct diagnostic strategies for each patient in a personalized way.

## Figures and Tables

**Figure 1 ijms-24-02449-f001:**
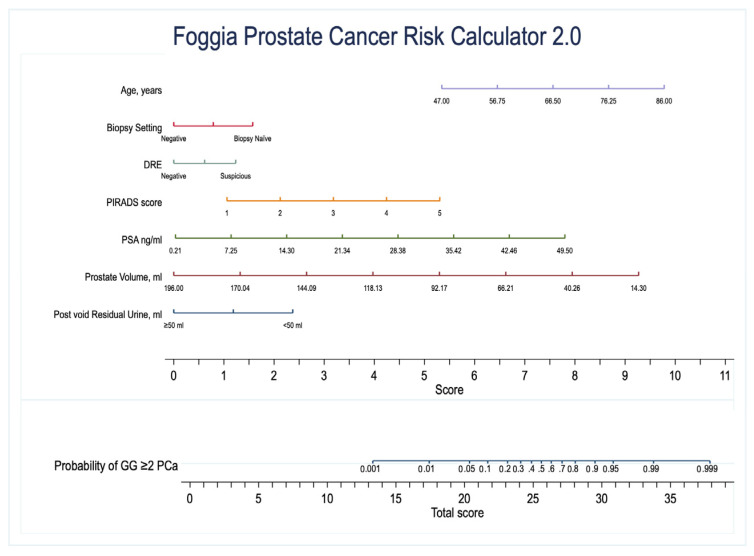
Nomogram to compute the predicted probability of csPCa (GG ≥ 2 PCa) using the Foggia Prostate Cancer Risk Calculator 2.0.

**Figure 2 ijms-24-02449-f002:**
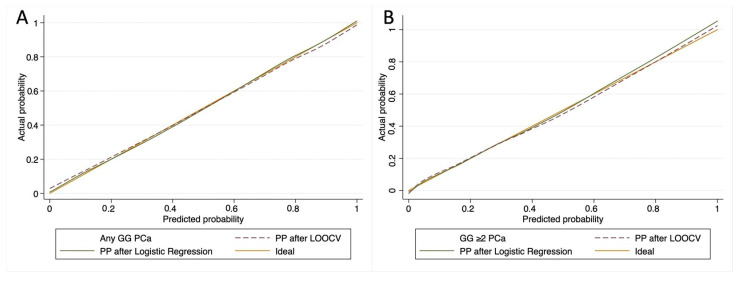
Calibration plot of observed vs. predicted probability of PCa (**A**) and csPCa (**B**) before and after leave-one-out cross validation.

**Figure 3 ijms-24-02449-f003:**
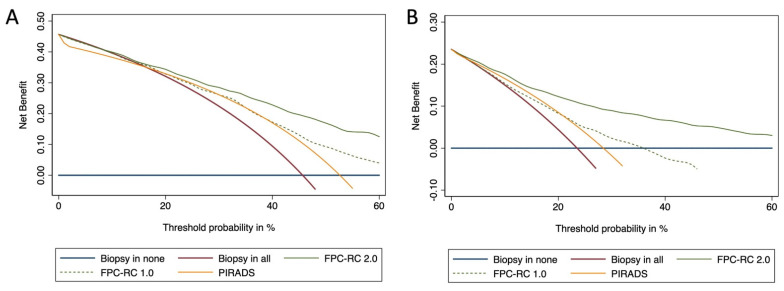
Decision curve analysis demonstrating net benefit between the threshold probabilities of 15% and 80% for the model predicting Any GG PCa (**A**) and at each threshold probabilities for the model predicting GG ≥ 2 PCa (**B**). The Foggia PCa Risk calculator 1.0 (without MRI) and 2.0 (with MRI) are compared.

**Table 1 ijms-24-02449-t001:** Descriptive Characteristics of Patients.

	Overall N = 728	Negative Bx (N = 395)	Bx ISUP GG 1 PCa (N = 161)	Bx ISUP GG ≥ 2 PCa (N = 172)	*p* Value
**Age, years**	66.5 (60.0, 71.0)	65.0 (59.0, 70.0)	66.6 (61.0, 71.0)	69.0 (64.0, 73.0)	**<0.0001**
**5-ARI, n (%)**					
No	659 (90.4%)	356 (90.1%)	151 (93.2%)	152 (88.4%)	0.3
Yes	70 (9.6%)	39 (9.9%)	11 (6.8%)	20 (11.6%)	
**PVR, n (%)**					
0–50	512 (70.2%)	252 (63.8%)	111 (68.5%)	149 (86.6%)	**<0.0001**
>50	217 (29.8%)	143 (36.2%)	51 (31.5%)	23 (13.4%)	
**PVR, mL**	30.0 (0.0, 50.0)	30.0 (0.0, 50.0)	20.0 (0.0, 50.0)	30.0 (0.0, 40.0)	**0.046**
**Q Max, n (%)**					
0–10	99 (13.6%)	63 (15.9%)	20 (12.3%)	16 (9.3%)	0.092
>10	630 (86.4%)	332 (84.1%)	142 (87.7%)	156 (90.7%)	
**Q Max, mL/s**	14.9 (10.8, 20.7)	14.2 (10.2, 20.0)	15.0 (11.0, 20.3)	15.8 (11.5, 28.0)	**0.037**
**IPSS**	9.0 (5.0, 15.0)	10.0 (5.0, 16.0)	7.0 (3.0, 12.0)	9.0 (5.0, 15.0)	**0.001**
**Biopsy History, n (%)**					
Biopsy naive	481 (66.0%)	230 (58.2%)	122 (75.3%)	129 (75.0%)	**<0.0001**
Previous Negative	248 (34.0%)	165 (41.8%)	40 (24.7%)	43 (25.0%)	
**DRE, n (%)**					
Negative	469 (64.3%)	281 (71.1%)	111 (68.5%)	77 (44.8%)	**<0.0001**
Suspicious	260 (35.7%)	114 (28.9%)	51 (31.5%)	95 (55.2%)	
**PSA, ng/mL**	6.1 (4.5, 9.0)	6.0 (4.4, 8.9)	6.0 (4.6, 8.1)	6.6 (4.6, 11.2)	**0.032**
**PSA density**	0.1 (0.1, 0.2)	0.1 (0.1, 0.2)	0.1 (0.1, 0.2)	0.2 (0.1, 0.3)	**<0.0001**
**Prostate volume, cc**	52.7 (39.6, 70.0)	60.0 (47.0, 79.0)	47.0 (38.0, 64.0)	40.0 (32.0, 53.7)	**<0.0001**
**PIRADS**					
1–2	140 (19.2%)	117 (29.6%)	19 (11.7%)	4 (2.3%)	**<0.0001**
3	185 (25.4%)	118 (29.9%)	40 (24.7%)	27 (15.7%)	
4	297 (40.7%)	122 (30.9%)	86 (53.1%)	89 (51.7%)	
5	107 (14.7%)	38 (9.6%)	17 (10.5%)	52 (30.2%)	

Continuous variables are reported as medians (interquartile range); categorical variables are reported as rates (n). DRE, digital rectal examination; ISUP GG, International Society of Urological Pathology Grade Group; PCa, prostate cancer; Q MAX, peak flow rate; PSA, prostate-specific antigen; PVR, post-void residual urinary volume. The bold values are the statistically significant differences.

**Table 2 ijms-24-02449-t002:** Multivariable binary logistic regression analysis testing the value of clinical variables in predicting any GG of PCa and clinically significant prostate cancer (csPCa) (GG ≥ 2).

	Multivariable Model Predicting Any PCa AUC = 0.80			Multivariable Model Predicting ISUP GG ≥ 2 AUC = 0.84		
Covariate	OR	95% CI	*p* > |z|	OR	95% CI	*p* > |z|
**Age**	1.05	1.03, 1.08	<0.001	1.07	1.04, 1.10	**<0.001**
**Biopsy History**						
Biopsy naive	Ref.			Ref.		
Previous Negative	0.40	0.27, 0.58	<0.001	0.40	0.25, 0.65	**<0.001**
**DRE**						
Negative	Ref.			Ref.		
Suspicious	1.35	0.95, 1.94	0.099	2.02	1.34, 3.06	**0.001**
**PIRADS**						
1–2	Ref.			Ref.		
3	3.12	1.76, 5.52	<0.001	5.43	1.78, 16.58	**0.003**
4	6.13	3.60, 10.46	<0.001	10.32	3.58, 29.76	**<0.001**
5	5.01	2.60, 9.67	<0.001	14.61	4.76, 44.81	**<0.001**
**PSA**	1.05	1.01, 1.08	0.005	1.09	1.05, 1.13	**<0.001**
**Prostate Volume**	0.98	0.97, 0.98	<0.001	0.98	0.97, 0.99	**<0.001**
**PVR**						
0–50	Ref.			Ref.		
>50	0.57	0.39, 0.84	0.004	0.27	0.15, 0.47	**<0.001**

DRE, digital rectal examination; ISUP GG, International Society of Urological Pathology Grade Group; PCa, prostate cancer; PSA, prostate-specific antigen; PVR, post-void residual urinary volume.

## Data Availability

Data sharing not applicable.

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
