# Peer review of "Foggia Prostate Cancer Risk Calculator 2.0: A Novel Risk Calculator including MRI and Bladder Outlet Obstruction Parameters to Reduce Unnecessary Biopsies"

_ijms, 2023, doi:10.3390/ijms24032449_

Round 1
Reviewer 1 Report
More experimental evidences are required to support the conclusions.
Author Response
Reply 1.1. We thank the reviewer for his/her comment. We hope the revised paper meets the required evidence to support the conclusions. Additionally, please consider that we followed point by point each step for the development of a model. The last step would be External validation, and we hope our colleagues from other institution will validate our findings in their cohorts. Thank you
Reviewer 2 Report
Thank you for submitting your work to our Journal.
First, I suggest you include a brief description of the initial RC, I see you mention references 7 and 8 but I guess it would be more convenient to have a few words in this paper as well and use the references only for more details.
I would recommend adding a few words about the different ways of obtaining biopsies, since this is a major factor in the overall detection rate. A recent paper worth citing about this can be found by searching the DOI: 10.11152/mu-2832.
I also recommend a few words about the potential risk of the RC advising against biopsy in a patient with PCa. You already discuss the implication of the patient in the final decision but the doctor is has to have the last word in my view.
I will gladly review a revised version of your paper.
Author Response
Reply 2.1. We thank the reviewer for his/her valuable comments. We agree that more information’s are needed on the previous version of the RC. Accordingly, we revised the introduction and the discussion section. Thank you for this comment.
Reply 2.2. Thank you for pointing out this publication. We added a comment on this in the revised manuscript. Please find in the discussion section our comment on the matter.
Reply 2.3. Thank you for the opportunity to discuss this point. We agree with the reviewer that the final decision should be individualized and doctor advice as well. Our tool help in both the case. Regarding the risk against biopsy in a patient with PCa, we performed a dedicated analysis. Decision curve analysis is indeed a suggesting to use the model for csPCa at each cut-off probabilities. Conversely, the model predicting any PCa should not be used for probabilities below 18%. In these cases, is indeed higher the chance of not suggesting biopsy in patients with PCa. In these cases however, the likelihood to diagnose GG 1 PCa is higher and should be discussed with the patients.Reviewer 3 Report
I would submit the view that both prostate gland volume and PVR are reflecting the same variable relating to gland volume. I would want to know if the authors looked at PSA density (PSAD) using the more accurate MRI-based volume determinant to see if PVR and prostate volume alone or together still had significant value. I suspect that they will not.
Also, I find many papers not citing the work of Lepor et al. on the importance of gland volume and so-called benign-related PSA. In other words, the larger the gland the more the contribution of benign epithelium to PSA. Therefore, the actual PSA related to PCa is relatively diminished.
Lepor H, Wang B, Shapiro E: Relationship between prostatic epithelial volume and serum prostate-specific antigen levels. Urology 44:199-205, 1994.
Also, patients with LUTS and increased PVR are also likely to have a greater contribution to PSA 2° inflammation. Was this evaluated?
In the discussion in the 3rd paragraph you use the abbreviation "ISR". What does this mean? Also, that sentence stating that PI-RADS improved the productivity from 0.76 to 0.78 for any GG PCa and from 0.80 to 0.82 for csPCa seems not worth the added expense and time allotments to do a mpMRI.
I also could not see where you made it clear if the prostate volume was done per transrectal ultrasound (TRUS) or MRI, the latter being shown to be more accurate. If not done by MRI, then the PSAD reading is of course affected, and that would diminish the value of PSAD in your risk calculator.
Author Response
Reply 3.1 Thank you for the opportunity to comment on this. We agree that PVR and P Volume are connected by the common link to BPH. However, the severity of symptoms and the presence of high volume of residual urine after voiding are also connected to the degree of bulging for the central zone and is not uncommon to evaluate patients with low prostate volumes but high PVR and severe LUTS. Regarding the addition of PSAD: we calculated it based on the volume derived from the MRI, we tried two models, one with PSA density and one with PSA and Volume as separate covariates. The latter showed the highest AUC and best calibration. In both the models, PVR showed independent additive predictive value. Thank you for allowing us to further discuss this point.
Reply 3.2. We thank the reviewer for pointing out this reference we were not aware of. Is indeed interesting to assess the association between PSA, P volume and risk of PCa. While we are not able to perform analysis on the transition/central zone volume since this was not included in our data collection form, we believe this is an extremely important point to support the use of RCs compared to PSA density and PIRADS alone. In our model we have different Beta coefficients of the logit function for PSA (positive value) and Prostate volume (negative value). This allows to have better prediction of PCa risk. Conversely, using their ratio (ie. PSAd), based on the reference suggested, could be viewed as an excessive simplification since PSAd is not supposed to have a linear effect on PCa prediction. Further studies are however needed to further study the association between PSA, Central zone and Transition zone volume. Thank you for this very useful comment.
Reply 3.3. We thank the reviewer for his/her comment. We recently showed that PSA density rather than PSA should be used to rule out the risk of high PSA due to inflammation or LUTS. Our risk calculator implements both these parameters, and this is the main reason why it works. Thank you for your comment and for the opportunity to further stress the importance of this parameters. Please find the updated text in the discussion section.
Reply 3.4. We thank the reviewer for this comment. We apologies for the typo (ISR-PVR) and we corrected it. Thank you. Regarding the added value of PI-RADS score, we revised the manuscript to add a comment on this. We would like to stress that even if the AUC have a slight increase in the AUC, MRI offer the unique opportunity to locate and target suspicious lesions. Additionally the AUC is a good method to assess overall discrimination of the model, however DCA is what we are actually more interested because it compute and graphically present the clinical effect (biopsy performed/ number of PCa missed) of doing biopsies above certain cutoffs.
Reply 3.5. MRI prostate volume was used in the model and to compute PSA density. Please see reply 3.2.
Reviewer 4 Report
The manuscript submitted by Falagario et al. presents a new PCa RC including MRI and bladder outlet obstruction parameters (BOOP). The present study demonstrated that the addition of the parameters of MRI and bladder outlet obstruction (BOO) to standard clinical variables improves the predictive accuracy of risk calculators for prostate cancer. This new risk calculator represents a free, simple, and useful tool in order to better estimate the risk of PCa diagnosis and consequently guide the correct diagnostic strategies for each patient in a personalized way. However, the discussion needs to be more detailed. It is too short. In addition, the figures should be provided in a higher resolution. A brief introductory figure about the currently reported (used) calculating method(s) should be included (if any).
Author Response
Reply 4.1. We thank the reviewer for his/her nice summary of our work. The discussion has been enriched and figures have been reprinted in high quality. Regarding the introductory figure about the RC, we changed figure 1. The current figure 1 is now Supplementary figure 1. The New figure 1 include a nomogram to compute the predicted probability of csPCa using the Foggia Prostate Cancer Risk Calculator 2.0.
Round 2
Reviewer 4 Report
The article can be accepted in its current format. No new comments are to be addressed. Good luck!